# Mapping the yields of lignocellulosic bioenergy crops from observations at the global scale

Wei Li[1,2], Philippe Ciais[2], Elke Stehfest[3], Detlef van Vuuren[3], Alexander Popp[4], Almut Arneth[5], Fulvio Di Fulvio[6], Jonathan Doelman[3], Florian Humpenöder[4], Anna Harper[7], Taejin Park[8], David Makowski[9,10], Petr Havlik[6], Michael Obersteiner[6], Jingmeng Wang[1], Andreas Krause[5,11], Wenfeng Liu[2]

[1]Ministry of Education Key Laboratory for Earth System Modeling, Department of Earth System Science, Tsinghua University, Beijing, 100084, China
[2]Laboratoire des Sciences du Climat et de l'Environnement, LSCE/IPSL, CEA-CNRS-UVSQ, Université Paris-Saclay, 91191 Gif-sur-Yvette, France
[3]Department of Climate, Air and Energy, Netherlands Environmental Assessment Agency (PBL), The Hague, The Netherlands
[4]Potsdam Institute for Climate Impact Research (PIK), Potsdam, Germany
[5]Karlsruhe Institute of Technology, Institute of Meteorology and Climate Research – Atmospheric Environmental Research (IMK-IFU), Garmisch-Partenkirchen, Germany
[6]International Institute for Applied Systems Analysis, Ecosystem Services and Management Program, Schlossplatz 1, A-2361, Laxenburg, Austria
[7]College of Engineering, Mathematics, and Physical Sciences, University of Exeter, Exeter EX4 4QF, UK. 2 College of Life and Environmental Sciences, University of Exeter, Exeter EX4 4QF, UK
[8]Department of Earth and Environment, Boston University, Boston, MA 02215, USA
[9]CIRED, CIRAD, 45 bis Avenue de la Belle Gabrielle, 94130 Nogent-sur-Marne, France
[10]UMR Agronomie, INRA, AgroParisTech, Université Paris-Saclay, ThivervalGrignon 78850, France
[11]TUM School of Life Sciences Weihenstephan, Technical University of Munich, Freising, Germany

*Correspondence to*: Wei Li (wli2019@tsinghua.edu.cn)

**Abstract.** Most scenarios from Integrated Assessment Models (IAMs) that project greenhouse gas emissions include the use of bioenergy as a means to reduce $CO_2$ emissions or even to achieving negative emissions (together with CCS). The potential amount of $CO_2$ that can be removed from the atmosphere depends, among others, on the yields of bioenergy crops, the land available to grow these crops and the efficiency with which $CO_2$ produced by combustion is captured. While bioenergy crop yields can be simulated by models, estimates of the spatial distribution of bioenergy yields under current technology based on a large number of observations are currently lacking. In this study, a random forest algorithm is used to upscale a bioenergy yield dataset of 3,963 observations covering *Miscanthu*s, switchgrass, eucalypt, poplar and willow using climatic and soil conditions as explanatory variables. The results are global yield maps of five important lignocellulosic bioenergy crops under current technology, climate and atmospheric $CO_2$ conditions at a 0.5° × 0.5° spatial resolution. We also provide a combined "best bioenergy crop" yield map by selecting the one of the five crop types with the highest yield in each of the grid cell, eucalypt and *Miscanthus* in most cases. The global median yield of the best crop is 16.3 t DM ha$^{-1}$ yr$^{-1}$. High yields mainly occur in the Amazon region and Southeast Asia. We further compare our empirically derived maps with yield maps used in three IAMs and find that the median yields in our maps are >50% higher than those in the IAM maps. Our estimates of gridded bioenergy crop yields can be used to provide bioenergy yields for IAMs, to evaluate land surface models, or to identify the most suitable lands for future bioenergy crop plantations. The 0.5° × 0.5° global maps for yields of different bioenergy crops and the best crop and for the best crop composition generated from this study can be download from https://doi.org/10.5281/zenodo.3274254 (Li, 2019).

## 1. Introduction

Bioenergy crops have for a number of years been promoted as a source of renewable energy under policies from the European Union and the U.S. (WBGU, 2009). They have also gained increasing attention as a global climate mitigation option (Berndes et al., 2003; Rose et al., 2014; van Vuuren et al., 2009). Bioenergy with carbon capture and storage (BECCS) is an important negative emissions technology being used by integrated assessment models (IAMs) to develop different climate mitigation scenarios (Fuss et al., 2018; Popp et al., 2017; Rogelj et al., 2018). BECCS contributes a

cumulative carbon-dioxide removal (CDR) between 150 and 1200 Gt $CO_2$ in different future scenarios that limit global warming at 1.5 °C in 2100 compared to the preindustrial period (Rogelj et al., 2018). This wide range of CDR is mainly caused by the different Shared Socio-economic Pathways (SSPs) used in IAMs as well as by different model settings (Popp et al., 2014; Rogelj et al., 2018).

Grain or high-sugar crops like maize and sugarcane based on first-generation conversion technologies are not frequently

considered by IAMs because of their lower energy yields, high fertilizer requirements and the increasing food demand pressure in future scenarios (Karp and Shield, 2008). Bioenergy production systems in IAMs thus often refer to lignocellulosic bioenergy and correspond to perennial grasses (e.g. switchgrass and *Miscanthus*) and/or fast-growing trees (e.g. poplar, willow and eucalypt) coupled with technologies for converting lignocellulosic biomass to bioenergy (second-generation) (Karp and Shield, 2008). They can grow under a wider range of climatic conditions and soil types and have a

lower demand for fertilizer (Cadoux et al., 2012; Miguez et al., 2008) and a larger greenhouse gas (GHG) abatement potential than first generation biofuels (El Akkari et al., 2018). However, the competition for land used to grow bioenergy crops and other land uses (e.g. food, timber, wild species protection) seems inevitable, causing direct and indirect land-use change (LUC) and carbon emissions (Robertson et al., 2017; Smith et al., 2016). One option for minimizing the land competition and the consequent LUC emissions is to plant lignocellulosic bioenergy crops on "marginal lands" (Robertson et

al., 2017). So-called marginal lands are mainly assumed to be abandoned lands that were formerly used for agriculture. Reasons for the agricultural land abandonment may include degraded soil quality, low crop price or environmental and ecological protection (Kang et al., 2013; Tang et al., 2010).

The biomass yields of bioenergy crops on marginal lands or in future land use scenarios simulated by IAMs are often estimated from small crop yield datasets (e.g. Cai et al., 2011; Havlík et al., 2011; Kyle et al., 2011; Tang et al., 2010) or

using a meta-analysis of experimental data extracted from scientific papers (Laurent et al., 2015). These approaches largely oversimplify the spatial variability of climatic conditions and soil properties. Alternatively, yields of bioenergy crops can be simulated by specific bioenergy crop models (e.g. Hastings et al., 2009; Miguez et al., 2009) or by dynamic global vegetation models (DGVMs) (Beringer et al., 2011; Li et al., 2018b). Specific bioenergy crop models represent physiological processes related to plant production and show a good performance of reproducing the biomass yields observations, but they

are semi-mechanistic models based on empirical relationships, and processes other than productivity (e.g. soil carbon dynamic) are largely not represented (Hastings et al., 2009; Miguez et al., 2008). In addition, they are often designed for only one or two bioenergy crop types. By contrast, the DGVMs use generic plant functional types (PFTs) to represent a group of plants with similar physiological and phenological characteristics and have complex processes representations related to the carbon cycle, i.e. photosynthesis, carbon allocation, respiration, phenology and soil carbon dynamics

(Guimberteau et al., 2018; Sitch et al., 2003). DGVMs with specific representation of bioenergy crops and calibrated using site-level data can provide global bioenergy crop yield maps, but it is difficult to perfectly match observed yields site-by-site, partly due to lack of explicit management information (e.g. genotypes, fertilization, plant density) in the DGVMs (Heck et al., 2016; Li et al., 2018b). Nevertheless, at least two IAMs (IMAGE and MAgPIE) use simulated bioenergy crop yield maps from the DGVM — LPJmL (Bonsch et al., 2016; Stehfest et al., 2014). Technological progress may be further considered in

IAMs for the future increase of bioenergy (and food) crop yields.

A detailed global map of bioenergy crop yields based on a large number of field observations that could be used to validate the model-based scenarios is currently lacking, to the best of our knowledge. Recently, global large datasets of second-generation bioenergy crop yields were compiled (LeBauer et al., 2018; Li et al., 2018a). These datasets provide observation-based crop yields as well as coordinates, climate conditions (e.g. temperature, precipitation), soil properties (e.g. clay fraction) and management information, which can potentially be scaled to the globe using Machine Learning algorithms. The derived global yield maps not only are valuable to estimate the global bioenergy production potentials but also can be used as input data to IAMs or to evaluate the performances of specific bioenergy crop models and DGVMs. Global yield maps could also help governments or companies identifying the most promising areas for growing bioenergy crops.

The objective of this study is thus to generate spatially explicit bioenergy yields with a machine learning algorithm (Random Forest, Breiman, 2001) trained from a global yield dataset (Li et al., 2018a) with climate, soil condition and remote sensing variables as explanatory variables. The bioenergy crop yield maps produced by the machine learning algorithm at a 0.5° × 0.5° spatial resolution are then compared with the yield maps previously used in three IAMs, i.e. IMAGE (Stehfest et al., 2014), MAgPIE (Popp et al., 2014) and GLOBIOM (Havlík et al., 2011).

## 2. Materials and methods

### 2.1 Data

The global yield dataset used here was compiled from 3,963 published field measurements of five main lignocellulosic bioenergy crops: eucalypt, *Miscanthus*, switchgrass, poplar and willow (Li et al., 2018a). All yield records have coordinates (latitude and longitude) and crop types. Other information was also documented if it was reported in the original publications, including mean annual temperature (MAT), mean annual precipitation (MAP), soil clay fraction (CF), planting information (e.g. density, rotation length, harvest time, age) and management practices (irrigation and fertilization). Most yield data in this dataset correspond to the mean annual harvested biomass (Li et al., 2018a), and only about one-third of yield data were reported with age (Li et al., 2018a), so age is not specifically used in this study since we aimed to produce a spatial yield map for present day without temporal variability. Only 36%, 51% and 14% of the yield observations were reported together with MAT, MAP and CF, respectively (Li et al., 2018a). For those sites without such information, we used climate data from the CRUNCEP gridded dataset (Viovy, 2011) and CF data from Harmonized World Soil Database (HWSD v1.2, Nachtergaele et al., 2012) (Table 1). For the sites with reported MAT and MAP in the yield dataset, we compared the reported values with MAT and MAP from CRUNCEP at the corresponding grid cell and they are in a good agreement (Fig. S1). But the consistency is low between CF from HWSD and those reported in the site-level yield dataset (Fig. S1), probably due to the limited number of observations and strong heterogeneity of soil properties.

In addition to MAT, MAP and CF, we also used other explanatory variables (Table 1): 1) shortwave radiation (SR) derived from the MODIS products (Ryu et al., 2018), 2) growing season length (GSL) calculated using daily temperature from CRUNCEP (Viovy, 2011), 3) a soil water availability index (WAI) calculated from a soil water balance model using ERA-interim reanalysis data as inputs (see details in Tramontana et al., 2016), and 4) growing season summed normalized difference vegetation index (NDVI) from the MODIS NDVI dataset (Park et al., 2016). GSL is defined as number of days between the first five successive days with daily average temperature greater than 5 °C and the first five days with daily temperature smaller than 5 °C in a year (Frich et al., 2002; Mueller et al., 2015). For this calculation the years was set to start on January 1 in the northern hemisphere and on July 1 in the southern hemisphere.

Because the spatial resolution of CRUNCEP and the WAI data is 0.5° × 0.5° (Table 1), we performed all analyses at this spatial resolution. Thus, for CF and SR datasets with higher resolutions (Table 1), the median values in each 0.5° × 0.5° grid cell were used as explanatory variables. Although NDVI covers non-bioenergy vegetation type, we used the maximum value in each 0.5° × 0.5° grid cell from the original 0.05° × 0.05° resolution as a spatial proxy of the maximum yield potential that

bioenergy crops can reach in the machine-learning upscaling model. The multi-year median values of MAT and MAP from CRUNCEP, SR, WAI and NDVI between 2001 and 2010 for each grid cell were used to eliminate temporal variability.

## 2.2 Random Forest modelling

### 2.2.1 Random Forest

Random Forest (RF) has been used to analyze the relationships between independent variables and explanatory variables (e.g. the relation between crop yields and climate by Hoffman et al., 2018) and for up-scaling local data (e.g. global soil carbon loss due to human land use by Sanderman et al., 2017). RF is a machine learning algorithm that combines a set of regression trees constructed from a random subset of the observations (Breiman, 2001). Because each tree fitting in the forest uses a bootstrap sample of the training observations, the part of the data set not used is called out-of-bag (OOB) and can be used to test the tree prediction. This helps RF to be fit and validated when being trained, and thus no extra independent validation dataset is needed.

Here we used Python scikit-learn module (Pedregosa and Varoquaux, 2011) to perform the RF regressions. We set the number of trees in forest to be 1000, and the maximum depth of each tree (branch levels) to be 10. We verified that the coefficient of determination ($R^2$) between predictions and observations in the training data, and $R^2$ of OOB validation remain constant with number of trees larger than 1000 or maximum depth larger than 10 (Fig. S2). The importance of a variable can also be calculated in the scikit-learn module based on how much each variable decreases the weighted impurity, i.e. the sum over the number of splits across all trees that include this variable, weighted by the number of samples it splits (Louppe et al., 2013). Although the RF model is robust to correlated explanatory variables, the importance calculation could be biased if there are a strong collinearity between different variables. We thus calculated the correlations between all continuous explanatory variables (Table 1) in the training dataset (see **Section 3.1**).

### 2.2.2 Model training

The workflow of RF training and predicting is shown in Fig. S3. The median yield, MAT, MAP and CF of all site observations for each crop type in each 0.5° × 0.5° grid cell from the global yield dataset were calculated to build the training set. That is, for example, several yield observations were reported in the same 0.5° × 0.5° grid cell, the median value of these observations was used for this grid cell. This gives a total of 273 0.5° × 0.5° grid cells with yield observations. The SR, GSL, WAI and NDVI in these grid cells that are not recorded in the yield observation dataset were derived from each corresponding dataset (Table 1) and added in the training set. Crop type (CT, Table 1) was taken as a categorical variable in the RF training and was thus converted to five dummy variables, i.e. CT_eucalypt, CT_Miscanthus, CT_poplar, CT_switchgrass and CT_willow. Taking one yield observation of eucalypt for example, CT_eucalypt was set to 1 and the other four CTs were set to 0. Alternatively, we also tried one RF regression for each individual crop type as a sensitivity test for this categorical variable (see **Section 4.2**).

We first trained the RF model using data from all the 273 grid cells. However, the OOB $R^2$ (0.29) is low, indicating the poor performance of the trained model. The low OOB $R^2$ is probably because part of observed yields cannot be explained by the spatially explicit climate and soil conditions used as the explanatory variables in the model training. For example, some strong genotypes may produce high yields under poor climate conditions while low yields may be observed at some sites with poor soil conditions that are not representative for the whole 0.5° × 0.5° grid cell. In order to derive the best RF model for prediction, therefore, we further adopted a leave-one-out method (Siewert, 2018; Tramontana et al., 2015). Specifically, RF models were trained each time by excluding one grid cell in the training set. The RF model was then used to predict the yield for this excluded grid cell. The comparison between observations and predictions is shown in Fig. S4. There are 112 grid cells with predicted yields that are biased more than 1-σ of the observed yields (gray dots in Fig. S4a). These strongly biased grid cells were masked, and the remaining 161 grid cells retained to train the RF model again to obtain the best RF

model. The predicted yields from the best RF model agrees well with the observations (Fig. S4b), and $R^2$ of the OOB validation is 0.63. Note that the OOB $R^2$ (0.63) serves as an evaluation of the RF model performance rather than the $R^2$ between predictions and observations in the training set (0.95, Fig. S4b). In the RF model training, one can always get a very high $R^2$ for the training set by expanding the tree depth, but in that case, the RF model will be overfitted and thus have a poor ability to predict, suggested by a low OOB $R^2$.

The spatial distribution of the selected grid cells for model training is shown in Fig. 1. There is a good observation coverage in the US, Europe, China, Southeast Brazil and South Australia but sites are sparse in other regions (Fig. 1). Eucalypt, *Miscanthus*, switchgrass, poplar and willow take 16.8%, 24.2%, 16.8%, 26.1 and 16.1% of the total number of selected sites in the training data (Fig. 1).

### 2.2.3 Model prediction

After training by data from the selected 161 grid cells, the derived RF model was used to predict the global distribution of bioenergy crops yields. Specifically, the gridded values of continuous explanatory variables on each 0.5° × 0.5° land grid cell were derived from data sources listed in Table 1. Five predictions were made, each with one individual prescribed bioenergy crop type (e.g. CT_eucalypt = 1 and the other four CTs = 0 for eucalypt).

Although there are some drought and/or cold tolerant *Eucalyptus* species, most species have a limited cold tolerance and relative high demands for water and are thus usually cultivated in tropical and warm temperate regions (Jacobs, 1981). Also, because the RF model has a poor ability in extrapolation when the values of explanatory variables are outside the ranges of training data, we only limited each crop predictions in the areas that are adequate for growth. Specifically, the minimum MAT and MAP over all grid cells in the training dataset were derived for each crop. The regions adequate for growth of each bioenergy crop were then defined as grid cells with MAT and MAP higher than the minimums in the training data. In another word, if either MAT or MAP in a grid cell is lower than the minimums where a crop type is grown in the training data, this grid cell is excluded for upscaling the yield of this crop. The grid cells with adequate growth conditions for each bioenergy crop type are shown in Fig. S5a-e. We also provided an integrated map (Fig. S5f) where at least one bioenergy crop type can grow to represent the grid cells that can have yield predictions.

Beyond the five predictions made for each bioenergy crop, we derived a prediction of the "best crop" by selecting the bioenergy crop with highest yield in each grid cell to indicate the maximum achievable yields (see **Section 3.2**).

### 2.3 Bioenergy crop yield maps in IAMs

We compared our derived yield maps from RF with the bio-energy yields from three IAMs: IMAGE (Stehfest et al., 2014), MAgPIE (Popp et al., 2014) and GLOBIOM (Havlík et al., 2011). The yields used in IMAGE and MAgPIE are simulated by a DGVM — LPJmL (Beringer et al., 2011) and have separate yield data for woody (representing poplar, willow and eucalypt) and herbaceous (representing switchgrass and *Miscanthus*) bioenergy crops. For comparison, we used the present day (2010) actual yield maps (derived from RF).

In the IMAGE integrated assessment model framework (Stehfest et al., 2014), the LPJmL model is an integral component for crop and grass yields, hydrology, dynamic vegetation and carbon dynamics (Müller et al., 2016). Bioenergy crop yields for sugarcane, maize, and herbaceous and woody crops are represented on the grid-level in LPJmL and represent potential yields under current technology. In the IMAGE-Land model, these potential yields are calibrated on the regional level to currently observed yields based on Gerssen-Gondelach et al. (2015) for the present day. Future projections of bioenergy crop yield depend on scenario-specific assumptions of technological progress (Daioglou et al., 2019), but the yield map used in this study is for year 2010 and without future yield improvements

The yield map for year 2010 from MAgPIE used for comparison in this study includes the yield improvements due to technological development from 1995 to 2010. In the yield maps used as an input to MAgPIE, the potential bioenergy crop

yields simulated by LPJmL (Beringer et al., 2011) were reduced using information about observed land-use intensity (Dietrich et al., 2012) and agricultural area (FAO, 2013) because MAgPIE aims to represent actual yields (Bonsch et al., 2016; Humpenöder et al., 2014). It is assumed that LPJmL bioenergy yields represent yields achieved under highest currently observed land use intensity, which is observed in Europe. Therefore, LPJmL bioenergy yields for all other regions than Europe are reduced proportional to the land use intensity in the given region. In addition, yields are calibrated at the regional level to meet FAO agricultural area in 1995, resulting in a further reduction of yields in all regions. MAgPIE bioenergy yields can exceed LPJmL bioenergy yields over time as endogenous investments in R&D (Research and Development) pushing the technology frontier.

The bioenergy crop yield map used in GLOBIOM represents the yields from short-rotation tree plantations (i.e. *Eucalyptus*, *Acacia*, *Gmelina*, *Betula*, *Populus*, *Salix*) and thus only woody bioenergy crops. To generate this map, field measured yields (i.e. as mean annual increments of stem wood) for different short-rotation tree species with proper managements were first collected from various databases (dated between 1984 and 2006 and sourced from different global regions) and then scaled up to a global yield map based on the spatial patterns of potential net primary productivity from Cramer et al. (1999). The estimation of area potentials for tree plantations in the GLOBIOM maps followed an approach similar to the one proposed by Zomer et al. (2008), including thresholds of tree growth based on aridity, temperature, elevation, population density, and existing land cover (Havlík et al., 2011).

## 3 Results

### 3.1 Explanatory variables importance

The importance of explanatory variables to the RF model is shown in Fig. 2a, indicating their contributions to the overall tree splits in the forest. We verified that spatial $R^2$ is generally low between any pair of variables (median $R^2$=0.06, Interquartile range, IQR=0.14) with a maximum $R^2$ of 0.6 between MAT and GSL (Fig. 2b).

MAP is the most important variable in the RF regression with a contribution of 18.0% to the overall tree splits. Another water related variable, WAI derived from a simple bucket model with rainfall and evapotranspiration (ET) datasets, also has a significant contribution (11.9%) but we note here that ET from observations over natural and cultivated systems may be different from ET in a world with large areas covered by bioenergy crops. The second important variable is GSL, contributing 17.5% to the tree splits. However, it should be noted that the correlation between GSL and MAT is relatively high ($R^2 = 0.6$, Fig. 2b) because GSL was calculated using daily temperature. The contributions of GSL and MAT (4.3%) may thus be not well separated because of the collinearity, but it did not influence the prediction because RF prediction is not sensitive to the collinearity of explanatory variables. Nevertheless, it implies that temperature related variables are also very important for predicting the bioenergy crop yields in addition to MAP and WAI. Overall, water-related and temperature-related variables (MAP, WAI, GSL and MAT) are the most important variables cumulating an importance level of 51.7%.

Among bioenergy crop type dummy variables used in the RF model, CT_eucalypt and CT_Miscanthus have marked contributions (14.8% and 10.8%) while the contributions from other crop types (CT_poplar, CT_switchgrass and CT_willow) are low (<3%, Fig. 2a). This reflects the fact that eucalypt and *Miscanthus* are generally more productive than others (Li et al., 2018a). The total importance of all bioenergy crop types indices is 29.6%.

NDVI, as a proxy of maximum plant productivity in each grid cell (**Section 2.1**), and shortwave radiation (SR) contribute 8.4% and 6.9% to the trained RF model. CF, as the only soil property used in the regression, has a minor contribution of 3.5% (Fig. 2a), indicating that soil conditions may have little impact on the bioenergy crop yields. However, this should be interpreted cautiously considering the mismatch between CF from HWSD dataset and from the yield observation dataset based on field measurements (Fig. S1c).

## 3.2 Predicted climate-limited yields

The spatially explicit yield maps of different bioenergy crops were predicted based on the climatic and soil conditions in each grid cell (Fig. 3). MAP is the most important variable in the RF regression (Fig. 2a), and thus the predictions largely depend on the spatial patterns of annual rainfall. This is consistent with previous studies that MAP is the main predictor of NPP across spatial gradients (Knapp et al., 2017). Although the general spatial patterns seem similar, there are still differences caused by other factors than MAP. In general, eucalypt and *Miscanthus* have higher yields than the other three bioenergy crops (poplar, willow and switchgrass). The global median yields of eucalypt and *Miscanthus* in the considered regions are 16.0 (4.1, IQR of grid cells adequate for growth, same below) and 15.3 (2.0) t DM (ton dry matter) ha$^{-1}$ yr$^{-1}$ (Fig. 3a,b). The spatial distributions of predicted yields for poplar, willow and switchgrass show a similar pattern (Fig. 3c-e) because of the low importance of these three crop types in the RF regression (Fig. 2a). Still, the global median yields are slightly different, i.e. 10.1 (1.7), 10.6 (1.7) and 10.3 (1.6) t DM ha$^{-1}$ yr$^{-1}$ for poplar, willow and switchgrass respectively, mainly due to the difference in areas that are adequate for growth (Fig. S5).

The global median yield of the best bioenergy crop is 16.3 (7.0) t DM ha$^{-1}$ yr$^{-1}$ with highest yields in the Amazon area and Southeast Asia (Fig. 3f). Consistent with the high yields of *Miscanthus* and eucalypt, they are the main compositions of the best crop globally, occupying 41.3% and 35.9% of the total grid cells that are adequate for bioenergy crop growth (Fig. 3g). Eucalypt dominates as being the best crop in the wet tropical regions while *Miscanthus* distributes dominantly in the dry tropical regions and the temperate regions. Willow is the best crop in only 21.2% of the total grid cells, mainly in the regions with more severe conditions where other crops are excluded for growth based on the MAT and MAP ranges. The fractions of poplar and switchgrass are very low (Fig. 3g), indicating that they are not as competitive as the other crops in term of yields. Maps of yield differences between eucalypt and *Miscanthus* and among the other three crops are shown in Fig S6. There are substantial differences between the yields of eucalypt and *Miscanthus*. The higher yields of eucalypt than *Miscanthus* in South America, East US, central Africa and southeast Asia and lower yields in other regions (Fig. S6a) can also be reflected by the best crop type in Fig. 3g. Because the contribution of crop types (poplar, switchgrass and willow) is low the trained random forest algorithm (CT_poplar, CT_switchgrass and CT_willow in Fig. 2a), the predicted yields in the regions where all three crops can grow are controlled by other mutual variables and thus similar. Therefore, the yield differences among these three crops are mainly caused by the different adequate regions for growth (Fig. S5) defined by the minimum MAT and MAP in the observation dataset. For example, the regions adequate for willow growth include some areas with lower MAP like western US, eastern Europe and Central Asia (Fig. S5) than for poplar and switchgrass.

## 3.3 Comparison with maps used in IAMs

The comparison of best bioenergy crop yields in our RF map with the maps used in IMAGE, MAgPIE and GLOBIOM is shown in Fig. 4. The best crop yields refer to the higher yields between woody and herbaceous crops in each grid cell for IMAGE and MAgPIE and the woody crop yields for GLOBIOM since only short-rotation trees were included in this model. Compared to the RF map, yields are generally lower in the maps used in IAMs (Fig. 4c,e,g) with global median differences of -7.0, -8.1 and -5.2 ton DM ha$^{-1}$ yr$^{-1}$ for IMAGE, MAgPIE and GLOBIOM, respectively. But yields from the IAM maps are higher than the RF map in some regions, e.g. Southeast US, Southeast Asia for the MAgPIE map (Fig. 4e) and some places in Brazil and North China for the GLOBIOM map (Fig. 4g). Much lower yields in the IAM maps than the RF map were found in the equatorial winter dry ("Aw" category based on Köppen−Geiger Climate Classification (Kottek et al., 2006)) regions in southeast Brazil, Africa, India and Australia (Fig. 4c,e,g), especially for IMAGE and MAgPIE. In the equatorial full humid ("Af") and monsoonal ("Am") regions in South America (mainly Amazon region) and Africa (around the DRC), the yield difference is small between the RF and IMAGE and GLOBIOM maps (Fig. 4c,g). In the "Af" and "Am" regions in Southeast Asia, however, yields are lower from GLOBIOM than from RF but similar between IMAGE and RF

(Fig. 4c,g). For MAgPIE, yields are systematically lower than those from RF in these tropical regions except Southeast Asia (Fig. 4e). On the other hand, yields from MAgPIE are closest to the RF yields in all the three IAMs maps in Europe.

We also showed the best crop yield distribution histograms from different maps (Fig. 5). Most areas in the RF map have a yield range from 15 to 20 t DM ha$^{-1}$ yr$^{-1}$, and other areas located in another two ranges: 5 to 13 and 20 to 24 t DM ha$^{-1}$ yr$^{-1}$.

By contrast, a large fraction of areas from the IAMs maps are associated with yields lower than 15 t DM ha$^{-1}$ yr$^{-1}$ (Fig. 5). This is consistent with the generally higher yields in the RF map than the IAM maps in Fig. 4. In fact, the median mean yield in the regions where yields are available in the four datasets (the overlapped regions between Fig. 4a,b,d,f) from RF is >50% higher than the median yields from IAM maps (80%, 83% and 59% for IMAGE, MAgPIE and GLOBIOM, respectively). The shapes of yield distributions among IAMs are also different (Fig. 5). There are more areas with yields below 7 and

above 20 t DM ha$^{-1}$ yr$^{-1}$ in the IMAGE and MAgPIE maps than the GLOBIOM map. This is also reflected by the higher IQR from IMAGE (IQR=9.1) and MAgPIE (8.7) than GLOBIOM (5.7 t DM ha$^{-1}$ yr$^{-1}$). Although both IMAGE and MAgPIE yield maps are based on LPJmL, there are slight differences due to the calibration of the original potential yields of LPJmL to actual yields. Compared to IMAGE, MAgPIE has more areas with yields below 12 DM ha$^{-1}$ yr$^{-1}$ but less areas with yields between 17 and 22 DM ha$^{-1}$ yr$^{-1}$ (Fig. 5).

Yields from the IAM maps were also compared directly with yields from field site observations (Fig. 6) that were used to train the RF model (Fig. S4b). Consistent with the global results (Fig. 4, 5), yields from the three IAM maps were lower at most sites (median difference = -4.5, -4.3 and -2.0 DM ha$^{-1}$ yr$^{-1}$ respectively, Fig. 6a-c). Yields from IMAGE are roughly consistent with the site observations for switchgrass but much lower for *Miscanthus* and eucalypt (Fig. 6a). In MAgPIE, herbaceous crops (*Miscanthus* and switchgrass) yields lie around the 1:1 line but woody crops (eucalypt, poplar and willow)

yields are generally lower than the site observations (Fig. 6b). Because the bioenergy crops in the GLOBIOM maps refer to short-rotation trees, the yields are similar to the field measurements of willow and poplar (also switchgrass), but much lower compared to the observed yields of *Miscanthus* and eucalypt (Fig. 6c).

In addition to the comparison of the best crop yields, we also showed the yields of woody and herbaceous crops in each dataset respectively (Fig. S7). Yields of woody bioenergy crops in the IAM maps are lower than those in the RF map,

especially for IMAGE and MAgPIE. By contrast, the herbaceous crop yields from IMAGE and MAgPIE are close to the RF yields in some regions like Amazon and Southeast Asia.

# 4 Discussion

## 4.1 Yield comparison with other studies

Our estimated global median yields (Fig. 3) are generally within the ranges summarized by Searle and Malins (2014) from

field measurements in the literature for five second-generation bioenergy crops: 0-51, 5-44, 0-35, 0-21 and 1-35 t DM ha$^{-1}$ yr$^{-1}$ respectively for eucalypt, *Miscanthus × giganteus*, poplar, willow and switchgrass. The yields from RF also agree with the yield ranges of several bioenergy crop species (e.g. *Miscanthus x giganteus*, *Panicum virgatum*, *Salix*, *Populus*) based on published yield data (Laurent et al., 2015).

For Miscanthus and switchgrass, there are only small-scale experimental plots in different regions and no large-scale

plantation, so no region- or country-scale inventory data are available for comparison. Most yield data at farm levels were already included in our observation yield dataset (see "Field_type" and "Field_size" in Table 2 in Li et al., 2018a).

For poplar, willow and eucalypt, we collected some inventory data of mean annual increment (MAI) for species of *eucalyptus*, *populus* and *salix* for each country (Table S1, extracted from Table 6a in FAO, 2006). The volume unit of MAI was converted to mass unit of yield based on the wood density of different tree types (Engineering ToolBox, 2004). The

main difficulty is however lack of spatially explicit data about where are plantations located in national-scale inventory data, preventing an accurate comparison with the RF predicted yields. Still, we derived the yield range in the whole country from

the RF predicted yield maps and compared with the yield range from the inventory data (FAO, 2006, Fig. S8). Most yield ranges from the inventory data overlapped with the ranges from RF maps (e.g. eucalypt and willow in Argentina) although the former is generally lower than the latter (Fig. S8). The higher minimum and maximum yields from RF could be caused partly by the exclusion of regions with MAP and MAT below the minimums from the observation dataset (to avoid out-of-range prediction). Especially, in some large countries, the inventory data may have plantations in some harsh climate and soils (e.g. most eucalypt plantations distribute in drier areas in the South Brazil). However, we must note that it is not a fair comparison without knowing the exact plantation locations in each country.

## 4.2 Sensitivity tests and uncertainties in the RF model

We trained RF models using climatic and soil variables and observed yields at a resolution of 0.5° × 0.5°. However, climate and soil conditions at the observation sites may not match the mean values in the corresponding half-degree grid cell. In addition, the number of observation sites in a grid cell may also influence the derived median yields in this grid cell because of the possible sampling biases (e.g. all observations concentrating in a very small place that is not representative for the whole grid cell). We thus tried to train the model at a resolution of 0.01° × 0.01° using high resolution MAP and MAT from WorldClim (Hijmans et al., 2005) but the OOB $R^2$ did not improve. We also tried using shortwave incoming radiation (SR) from CRUNCEP (Viovy, 2011) instead of from Ryu et al. (2018). SR from CRUNCEP was simply converted from the cloudiness provided by CRU based on the calculation of clear sky incoming solar radiation as a function of date and latitude of each pixel (Viovy, 2011). By contrast, SR data from BESS was computed based on a series of forcing data from Terra & Aqua/MODIS Atmosphere and Land products, including solar zenith angle, dark target and deep blue combined aerosol optical depth, cloud optical thickness, cloud top pressure, cloud top temperature, surface pressure and surface temperature, total column precipitable water vapor and total ozone burden, and land surface shortwave albedo (Ryu et al., 2018). The SR data from BESS was also highly consistent with the observational field data ($R^2$=0.95, see Fig. 2 in Ryu et al., 2018). We still tested the RF performance using SR from CRUNCEP and the OOB $R^2$ remained unchanged (0.63), possibly due to the relatively low contribution of SR in the random forest training (Fig. 2a) and the high spatial correlation between SR from BESS and from CRUNCEP.

We also tried growing season integrated climate variables instead of annual mean values, but there is no significant improvement on the model training. Therefore, we focused our analyses on 0.5° × 0.5° grid cells using mostly mean annual values since the yield dataset only reported MAP and MAT (no growing season integrated values) from observations. In addition, the soil properties from HWSD are also highly uncertain (Fig. S1c) and the coarse resolution may not be able to represent the local soil conditions, partly explaining the low importance of CF in the RF model (Fig. 2). More detailed local soil property maps could help to improve the CF importance and thus the corresponding RF model performance.

We replaced the model-derived WAI with satellite-based surface soil moisture (SM) data, including the mean annual soil moisture data from Soil Moisture and Ocean Salinity (SMOS) during 2010-2018 (Li et al., 2020) and Soil Moisture Active Passive (SMAP) during 2015-2018 (O'Neill et al., 2019). The OOB $R^2$ for SMOS and SMAP are 0.60 and 0.59 respectively, compared to the original value of 0.63. The lower performance may be caused by the fact that satellite-based soil moisture data only accounted for soil water status in the top centimeters whereas productivity is influenced by root-zone soil moisture. In addition, the importance ranking changed from #4 for WAI (Fig. 2a) to #8 for SM_SMOS and SM_SMAP (Fig. S9). The relative order of other variables remains unchanged.

Although the total number of 0.5° × 0.5° grid cells (161) for RF training is relatively small compared to the global total land grid cells (> 60,000), However, the spatial representativeness of the sample is more important when being used to upscale the whole population pattern. As shown in Fig. S10, our training sample (gray) covers most ranges of climate and soil variables in the regions that we predicted (pink), implying that our training data are representative of the global adequate regions for bioenergy crop growth and thus appropriate for up-scaling. In addition to the range, the distributions also match

well between the training sample and the prediction region. Although the distributions of shortwave radiation are different, the importance of this variable in the RF model is low (Fig. 2a). Furthermore, to avoid possible biases induced by out-of-range prediction, we only limited our predictions in regions with MAT and MAP above the minimums in the training data (Section 2.2.3). Thus, this gives us 33,216 grid cells in the prediction regions (instead of >60,000 globally) and avoids biased predictions in regions that are beyond the capacity of our trained random forest model. At last, we would like to emphasize that the bioenergy crop yield observations were searched in published articles or reports in several literature databases and systematically collected (Li et al., 2018a), so it is impossible to include more grid cells (currently 273 half-degree grid cells, 161 after selecting) as there are no more observations available. Using these data, the OOB $R^2$ that serves as an evaluation of the trained random forest is 0.63, implying the trained RF algorithm is acceptable for prediction.

The temporal resolution and coverage of the training dataset are important for training the machine learning model given the temporal variations of climate conditions. Therefore, we analyzed the sampling time in the training dataset. There are ~30% of the yield observations without reported sampling year in the original dataset and also ~30% in the aggregated 0.5° × 0.5° data used for random forest training. We thus arbitrarily set the 2 years before the publication year as the sampling year for the yield observations without reported sampling years (e.g. set 1997 as the sampling year if the reference paper was published in 1999). The frequency of the sampling years in the 0.5-degree data used for random forest training is shown in Fig. S11. The sampling years range from 1969 to 2016 with a median year of 1999. We then derived temperature (T), precipitation (P) and short-wave radiation (SW, from CRUNCEP because BESS SW starts from 2001) and soil water availability index (WAI) at the sampling year for each grid cell and re-trained the random forest (RF). However, the OOB $R^2$ is 0.54, lower than the original value of 0.63. Possible reasons may include: 1) RF training may largely respond to the spatial gradients of climate and soil conditions, and thus the contribution of temporal variation may be low; 2) Climate conditions at the sampling year may be a good predictor of yields for annually harvested herbaceous crops, but yields of woody crops like eucalypt, poplar and willow may also be impacted by the previous years in the whole growing cycle. Unfortunately, there are only about 18% observations with both reported harvest year and age, impeding the derivation of the mean climate conditions during the whole growing cycle. In addition, using the climate conditions at the sampling years also changed the variable importance (Fig. S12) compared to the original one (Fig. 2a). Precipitation is no longer an important contributing variable while contributions of the other variables are more or less similar to those in the original trained RF.

Management factors like fertilization, irrigation, species and harvest time are important for bioenergy crop growth and impact the yields (Karp and Shield, 2008; Miguez et al., 2008). In the RF model training and prediction, however, we only used spatially explicit climatic conditions, clay fraction and crop type as explanatory variables, and other factors (e.g. management drivers) were not included because these explanatory variables are not available on a gridded basis. This may partly be responsible for the moderate OOB $R^2$ (0.63) in the model training. One other reason for the difficulty in taking management into account is the incomplete information reported for this variable from field measurements and thus in the yield observation dataset (Li et al., 2018a). For example, 75% of the observations did not report irrigation information (Li et al., 2018a). Another reason is that different management practices are difficult to harmonize. For example, fertilization may be applied annually, only one-time at plantation or irregularly (Li et al., 2018a); There are not enough data samples further classifying crop types (e.g. species or genotypes). Specific for the resolution of our analyses, it is difficult to derive a median or mean management quantification for a half degree grid cell from all observations inside. In addition, Crop age is an important factor in predicting the yields because of the growth cycle of perennial crops like *Miscanthus* (Lesur et al., 2013). However, the yield data in the observation datasets mainly refer to mean annual biomass yield which blended the growth cycle, especially for the trees (Li et al., 2018a). In the field measurements studies, biomass yield for trees is often calculated by the total biomass divided by age although some studies may report the biomass increment at a certain age. Also, because there are only about one-third observations with age information and we only aimed to produce a spatially explicit map, age is not used as an explanatory variable in the RF model.

We attempted a RF model training by including irrigation flag (yes or no), fertilization flag (yes or no) and/or fertilization frequency (annual or one-time). However, these attempts failed to improve the model and the importance of these factors was very low (<1%). Nitrogen application rate reported in the yield observation dataset was also taken as a continuous variable in the exploring RF model training, but it only contributed <4% to the total tree splits. Reasons for the low contribution of fertilization (flag, frequency, or application rate) may include unknown basic nutrient availability from soils, possible existence of nitrogen-fixing bacteria, and dry and wet nitrogen depositions. In addition, the yield response of *Miscanthus* to fertilizer application may be not significant (Cadoux et al., 2012; Heaton et al., 2004).

We took the bioenergy crop type (CT in Table 1) as a categorical variable in our RF model to include yield data from all crops in order to make a full use of the climate gradient information in the upscaling. However, this mixes climate information from one crop with the other crops and may induce some uncertainties. We thus trained one RF model for one individual bioenergy crop, and the OOB $R^2$ is 0.42, 0.02, 0.43, 0.19 and 0.42 for eucalypt, *Miscanthus*, poplar, willow and switchgrass, respectively. The OOB $R^2$ for individual crops is lower than the OOB $R^2$ of the original RF model using all crops (OOB $R^2$=0.63, **Section 2.2.2**), especially for *Miscanthus* and willow, probably because of the limited number of observations. Still, we mapped the yields for each individual crop with an OOB $R^2$ greater than 0.4 (i.e., eucalypt, poplar and willow) and compared with our original estimates (Fig. S13). Although there are some small differences for poplar and switchgrass, it barely influences our best crop results since poplar and switchgrass are the highest-yielding crops in only 1.6% of the cells (Fig. 3g). For eucalypt, our original estimates are higher than the yield predictions from the individual crop (eucalypt) RF model in Northwest Brazil and Southeast Asia but lower in other regions in Brazil and in the temperate regions (Fig. S13). The overall relative differences, however, are small for eucalypt with median positive and negative values of 4.0 (IQR=11.0) % and -7.2 (5.6) % respectively.

The prediction of RF model tends to be not reliable for predictors out of the training data range, and such extrapolation should be considered as inaccurate. We thus compared the distributions of variables in the training data and the global data used for prediction and provided the ranges for each bioenergy crop type in the training data (Fig. 10). Because we limited our predictions in the regions that are adequate for bioenergy crop growth using minimum MAT and MAP from observations (**Section 2.2.3**), the distributions of variables used for predictions largely overlapped the distributions from the training data (Fig. S10), implying that most of the predictions are reliable without extrapolations out of ranges. Although SR from 20 to 25 MJ m$^{-2}$ d$^{-1}$ is not presented in the training data (Fig. S10), the importance of SR in the RF model is relatively low (Fig. 2a), and thus the influence on our predictions is expected to be small. We should note that only minimum MAT is used to define the adequate regions, but some high temperature stress (e.g. through heat, vapor pressure deficit or summer drought) could also limit the growth. Although this is not explicitly considered in this study, the area with MAT higher than the maximum MAT in the training data is very small (Fig. S10).

Our predictions are based on current climate and $CO_2$ level, and thus the future climate changes and $CO_2$ fertilization effects are not included. The photosynthetic pathway for C4 plants (such as *Miscanthus* and switchgrass) is closer to optimal levels of $CO_2$ with present-day atmospheric levels. The $CO_2$ effect could result in large increases in productivity especially for the C3 plants, but data on bioenergy crop yield responses to $CO_2$ is very sparse, although this is being addressed in current field studies (e.g. Norby et al., 2016). We adopted a space-for-time approach and analyzed the spatial relationship between yields and temperature (Fig. S14) to account for the possible yield changes in response to future temperature changes due to adaptation. Yields are positively correlated to temperature for all bioenergy crops (Fig. S14, correlations with other explanatory variables are shown in Fig. S15-20). *Miscanthus* has the strongest response to temperature with an increasing rate of 0.41 t DM ha$^{-1}$ yr$^{-1}$ per °C. Eucalypt and willow have a similar increasing rate (0.27 and 0.26 t DM ha$^{-1}$ yr$^{-1}$ per °C). The temperature sensitivities of yields are lower for poplar and switchgrass (0.14 and 0.18 t DM ha$^{-1}$ yr$^{-1}$ per °C). The overall yield response to temperature for the best crop is 0.42 t DM ha$^{-1}$ yr$^{-1}$ per °C (Fig. S14). It is higher than each individual crop because it combined the yield gradient from multiple crops, so the yield sensitivities to temperature for the best crop also

comprise possible transitions of the low-yield crop type to the high-yield type. Based on an increase of 0.9 °C from the pre-industrial period until now (Millar et al., 2017), the temperature sensitivity of the best crop implies a mean increase of 0.46 t DM ha$^{-1}$ yr$^{-1}$ in yields from present days to 2100 in the 2 °C temperature increase scenario. However, this is just a simple extrapolation based on spatial gradients and should be interpreted cautiously. For example, future increase of soil aridity could cause soil degradations and counteract the yield increases due to $CO_2$ fertilization and temperature increase (Balkovič

et al., 2018).

## 4.3 Comparison with other yield maps

One potential application of our RF yield maps is to be used as an input to IAMs, so we made detailed comparisons with the currently used yields maps in three IAMs in terms of spatial patterns (Fig. 4), yield distributions (Fig. 5) and site-level yields (Fig. 6). Yields from the IAM maps are generally lower than those from our derived RF maps (Fig. 4,5) and the site-level

field observations (Fig. 6). One possible reason is that the IMAGE and MAgPIE models calibrated the simulated potential yields of LPJmL (highest yield that can be achieved by the best managements currently available) to the actual yields (see **Section 2.3**). The field observations are usually under some degree of management like irrigation or fertilization, and thus close to the potential yields, so IAMs reduced the yields using a calibration factor to represent the gap between the potential and actual yields, as the potential yields may not be reached in reality, especially in some low-income countries. As another

consequence of using data from well-managed field trials, the predicted yields from the RF model could be higher than the practical yields in large-scale plantations. Most of the observations in the training data are from small-scale experimental trials with managements rather than real farmers' fields (Li et al., 2018a). In addition, some yield observations are based on harvests at the peak yield time in summer or autumn rather than in winter or early spring after leaf falling and drying in practice. In fact, Searle and Malins (2014) reviewed bioenergy crop yields in the literature and concluded significantly lower

yields in semi-commercial scale trails than small plots because of the biomass drying loss and inefficient mechanical harvest. Crops in small plots may also benefit from the edge effect by receiving more light (Searle and Malins, 2014). But we should note that the median yield in each grid cell with multiple observations is used to the train the RF model, and thus some extremely high yield observations due to intensive managements may not contribute strongly to the trained RF model.

In addition, inclusion of or more dependence on the high-yield bioenergy crop types (i.e. *Miscanthus* and eucalypt) in the RF

model would also lead to higher yield predictions. For example, in LPJmL where the IMAGE and MAgPIE yield maps come from, switchgrass and *Miscanthus* were treated as one single PFT (Beringer et al., 2011; Heck et al., 2016) although these two crop types have very different physiological parameters and thus significant difference in yields (Dohleman et al., 2009; Heaton et al., 2008; Li et al., 2018b). The calibration of this one single PFT using both yields data from switchgrass and *Miscanthus* would overestimate yields of the former and underestimate the latter. For eucalypts, LPJmL seemed to

underestimate the yields in the first place (see the comparison with field measurements in Fig. 1b in Heck et al. (2016)). The RF model trained in this study, on the other hand, relies more on the crop types of *Miscanthus* and eucalypt (see their importance in Fig. 2a). Although yield maps from IMAGE and MAgPIE were both based on the LPJmL simulations, they showed some differences in spatial patterns, yield distributions and site-level yield comparison, due to different calibration processes for the yield data simulated by LPJmL (see **Section 2.3**).

Yields from the GLOBIOM map are close to the site-level observations of willow, poplar and switchgrass (Fig. 6c). Therefore, the lower yields from GLOBIOM than the best crop yields from RF is mainly caused by the inclusion of *Miscanthus* and more eucalypt observation data in the RF model. More contributions from these high-yield crops drive the yields higher in the RF predictions.

Accurate input data of bioenergy crop yields are crucial for IAMs to simulate the future land-use change through the trade-

off between BECCS and other climate mitigation options. The global median yields from our RF map are >50% higher than those used in IAMs in the overlapped regions (Fig. 5). Therefore, if our RF yield data are used in IAMs and all the other

conditions being equal, it will make the BECCS option more competitive and require less land for bioenergy crop plantation to achieve the same mitigation target, although gaps between predicted yields from RF and actual yields particular in low-income countries need to be further taken into account. Also, it may need more water and nutrients in order to sustain the

high yields. Although the yield response to fertilizers may be not obvious (Cadoux et al., 2012; Miguez et al., 2008), the net nutrient loss from biomass harvest must be replenished to maintain the nutrient balance in the soil and support further growth. In addition, we compared the yield map derived from random forest with the yields simulated by the land surface model — ORCHIDEE (Fig. S21). Because poplar and willow were taken as one plant functional type (PFT) in ORCHIDEE (Li et al., 2018b), the average yields of poplar and willow from random forest were used for comparison (Fig. S21b). The yields

simulated by ORCHIDEE are generally higher than those from random forest, especially for Miscanthus and Poplar&willow. This could be largely expected because in this version of ORCHIDEE, there are no nutrient limitations on plant growth, no effect of pests and disease on crops, and the management practices were implicitly included when adjusting the productivity parameters in the model to match the site observations with management like irrigation, fertilization or specific high-productive genotype. There could be a similar case in LPJml (Heck et al., 2016), and also why the IAMs calibrated the

LPJml yields based on currently observed yields to get the potential yield maps (Section 2.3). On the other hand, the predictions from random forest are largely constrained by the yield range of observations, representing the yields that can be achieved (or were achieved during the period when yield data were reported) under current (optimal) technology. This is exactly the purpose of producing this data product in our study, which is observation-based and can be used to benchmark the yields simulated by land surface models or IAMs.

**5 Data availability**

The field observed site-level yield data for major lignocellulosic bioenergy crops can be downloaded through https://doi.org/10.6084/m9.figshare.c.3951967 (Li et al., 2018a). The 0.5° × 0.5° gridded global maps for yields of different bioenergy crops and the best crop and for the best crop composition generated from the random forest model in this study can be download through https://doi.org/10.5281/zenodo.3274254 (Li, 2019).

**6 Conclusion**

We mapped bioenergy crop yields at the global scale using a machine learning method trained on field yield data and based on several climatic and soil conditions. In addition to evaluating the performances of IAMs and DGVMs, our spatially explicit bioenergy crop yields can also be used to determine the suitable lands with proper bioenergy crop yields, conduct life cycle assessment and estimate the nutrient removal from biomass harvest. Although there are a large number of field

measurements in the yield observation dataset used to build the RF model, the geographic coverage is poor in some regions. Therefore, more field measurements in regions with limited observations (e.g. Africa) and a proper quantification and synthesis of management factors will be useful to improve the predictions of global yields in future.

**Author contribution**

P.C. and W.L. conceived the study. W.L. analysed the data and drafted the manuscript. E.S., A. P., F.D.F., J.D., F.H., P.H. and M.O. provided the yield maps from IAMs. T.P provided the NDVI data. All authors contributed to the interpretation of the results and to the manuscript.

**Competing interests.** The authors declare that they have no conflict of interest.

**Acknowledgements**

This study is supported by the National Key R&D Program of China (grant number: 2019YFA0606604). W.L. was supported by the European Commission-funded project LUC4C (grant no. 603542). W.L. and P.C. are supported by the European Research Council through Synergy grant ERC-2013-SyG-610028 "IMBALANCE-P", and P.C. acknowledges support by the CLAND convergence institute of the French Agence Nationale de la recherche under the "Investissements d'avenir" programme with the reference ANR-16-CONV-0003.

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

**Table 1 Variables used in the upscaling of bioenergy crop yields.**

| Variable | Description | Original resolution | Data Source |
|---|---|---|---|
| CT | Crop type: eucalypt, *Miscanthus*, switchgrass, poplar or willow | - | Li et al. (2018a) |
| MAT | Mean annual temperature (°C) | $0.5° \times 0.5°$ | Li et al. (2018a); CRUNCEP (Viovy, 2011) |
| MAP | Mean annual precipitation (mm yr$^{-1}$) | $0.5° \times 0.5°$ | Li et al. (2018a); CRUNCEP (Viovy, 2011) |
| CF | Clay fraction | $30'' \times 30''$ | HWSD (Nachtergaele et al., 2012) |
| SR | Shortwave radiation (M J m$^{-2}$ d$^{-1}$) | $0.05° \times 0.05°$ | Ryu et al. (2018) |
| GSL | Growing season length (d) | $0.5° \times 0.5°$ | based on CRUNCEP (Viovy, 2011) |
| WAI | Soil water availability index | $0.5° \times 0.5°$ | Tramontana et al., (2016) |
| NDVI | Growing season summed normalized difference vegetation index | $0.05° \times 0.05°$ | Park et al. (2016) |

730

735

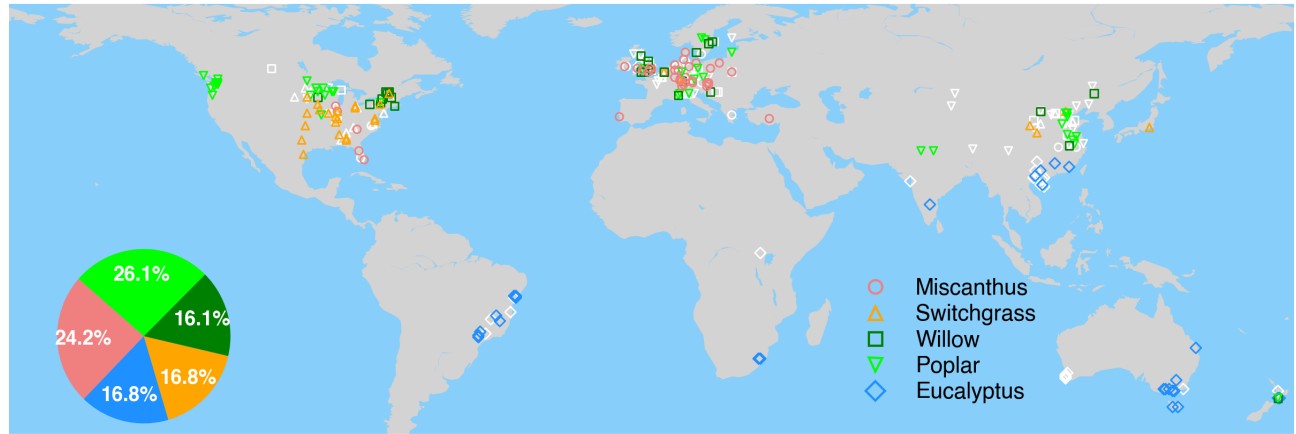

**Figure 1: Map of grid cells with yield observations in the global yield dataset. The colored and white markers indicate the selected (blue dots in Fig. S4a) and masked (gray dots in Fig. S4b) grid cells, respectively, based on a bias threshold of 1-σ for the RF modeling of these yields. The inset pie plot shows the percentages of each bioenergy crop types in the selected grid cells (colored markers) for model training.**

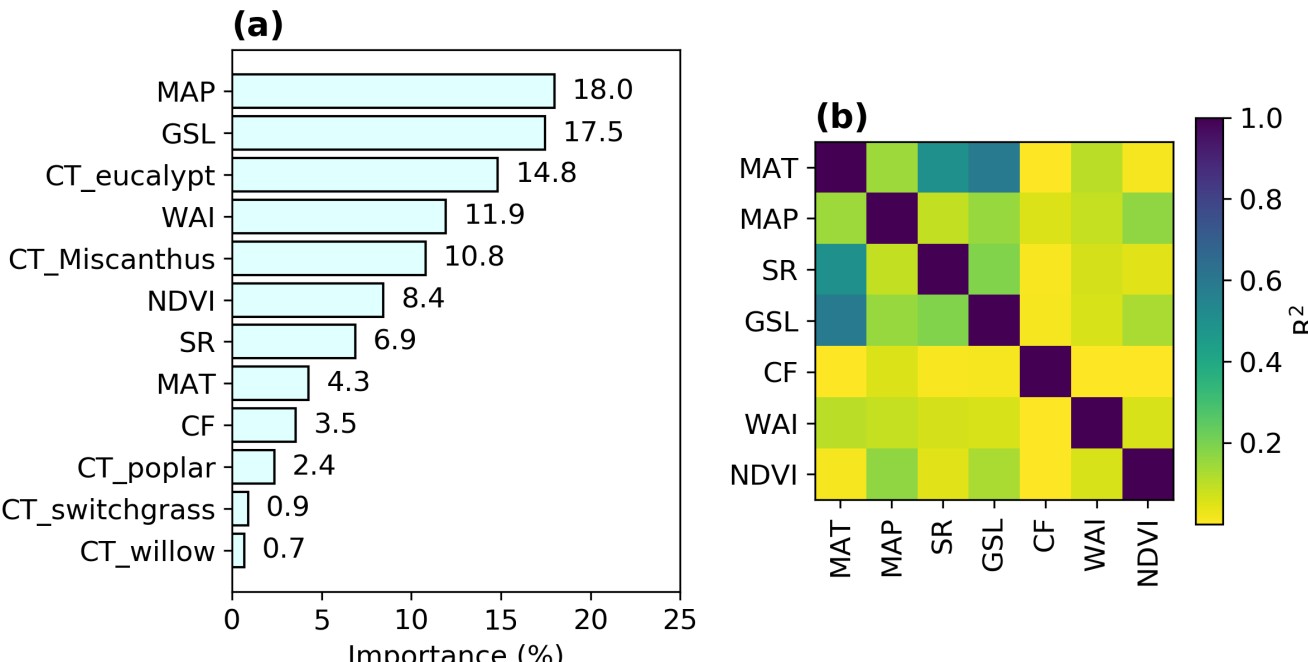

**Figure 2: Variable importance in the trained RF model (a) and $R^2$ from the regressions between different explanatory variables (Table 1) in the training data (b). The importance of one variable is calculated based on the sum over the number of splits across all trees that include this variable, weighted by the number of samples it splits. The relative contributions of each explanatory variable (summed to 100%) are shown here.**

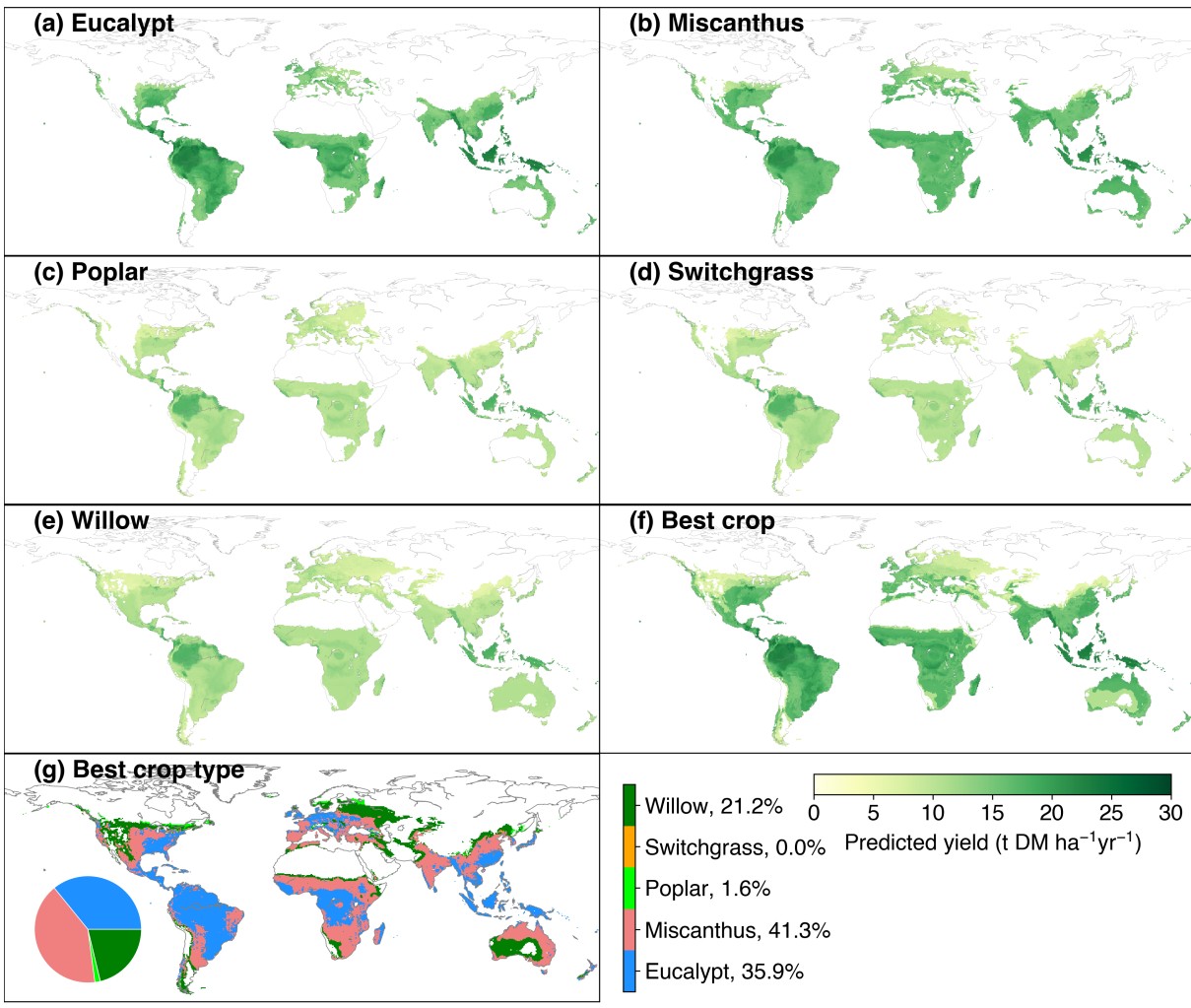

**Figure 3: Spatial distribution of predicted yields for different bioenergy crops (a-f) and best crop type in each grid cells that are adequate for growth (g). The inset pie plot in (g) shows the fractions of grid cells occupied by each bioenergy crop type. The white areas indicate regions where no prediction was derived due to inadequate conditions defined by minimum temperature and**
 **precipitation (see Methods).**

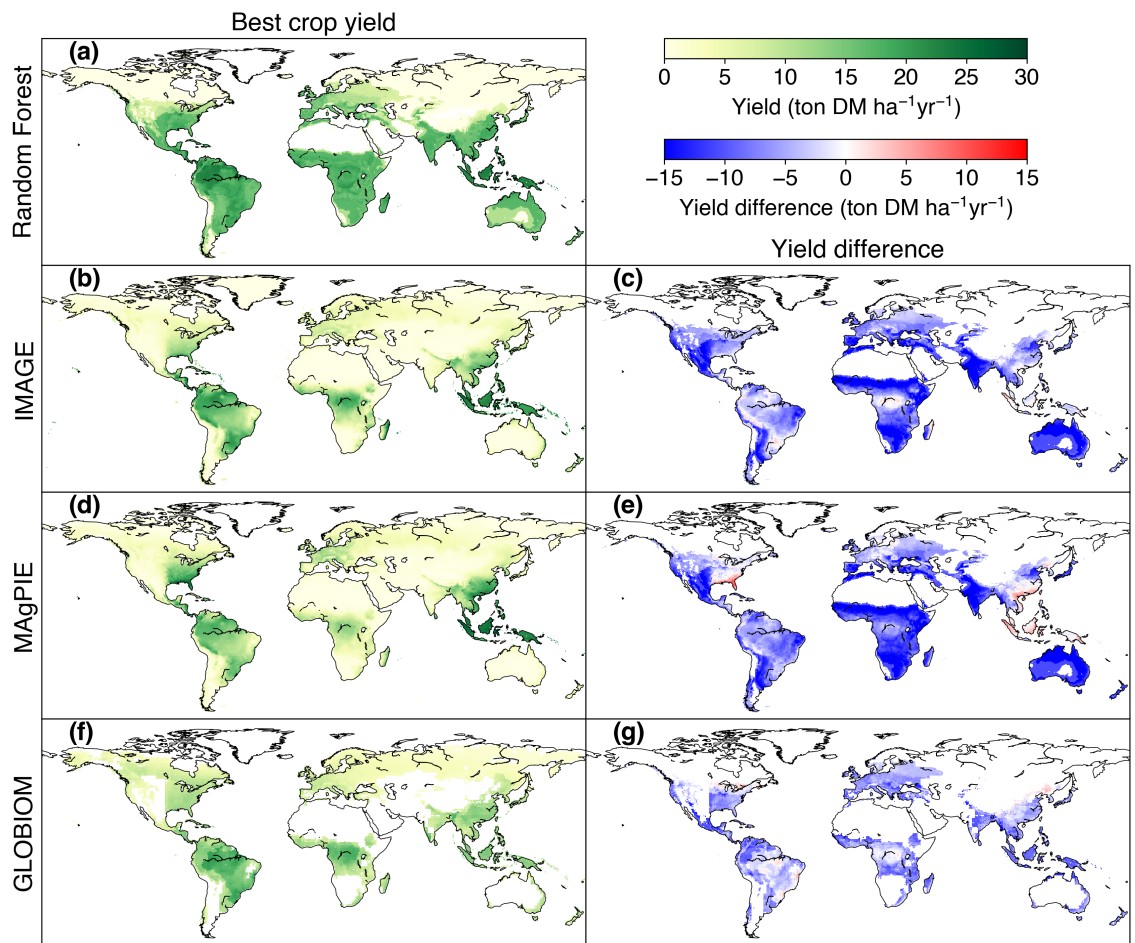

**Figure 4: Comparison of bioenergy crop yields between the RF map and maps used in three IAMs (IMAGE, MAgPIE and GLOBIOM).** The left panel (a, b, d, f) is the best crop yields from each dataset, and the right panel (c, e, g) refers to the yield differences between RF and each IAM maps (IAM yields minus RF yields where yields are available in both paired maps). The best crop yield map from RF (a) is the same as Fig. 3f. The best crop yields in IMAGE and MAgPIE (b, d) are the higher yields between woody and herbaceous bioenergy crops in each grid cell. The best crop yields in GLOBIOM (f) are the yields of woody crops (short-rotation trees) since there is no herbaceous bioenergy crop in GLOBIOM.

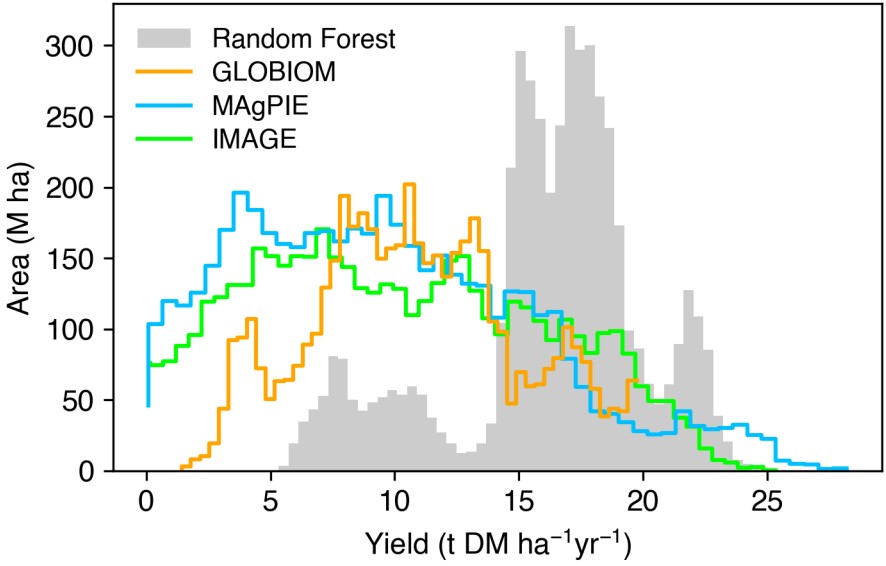

**Figure 5: Histograms of best crop yields in our RF yield map and yield maps used in the three IAMs. Only regions where yield values are available in all the four maps are used to generate the histogram.**

780

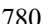

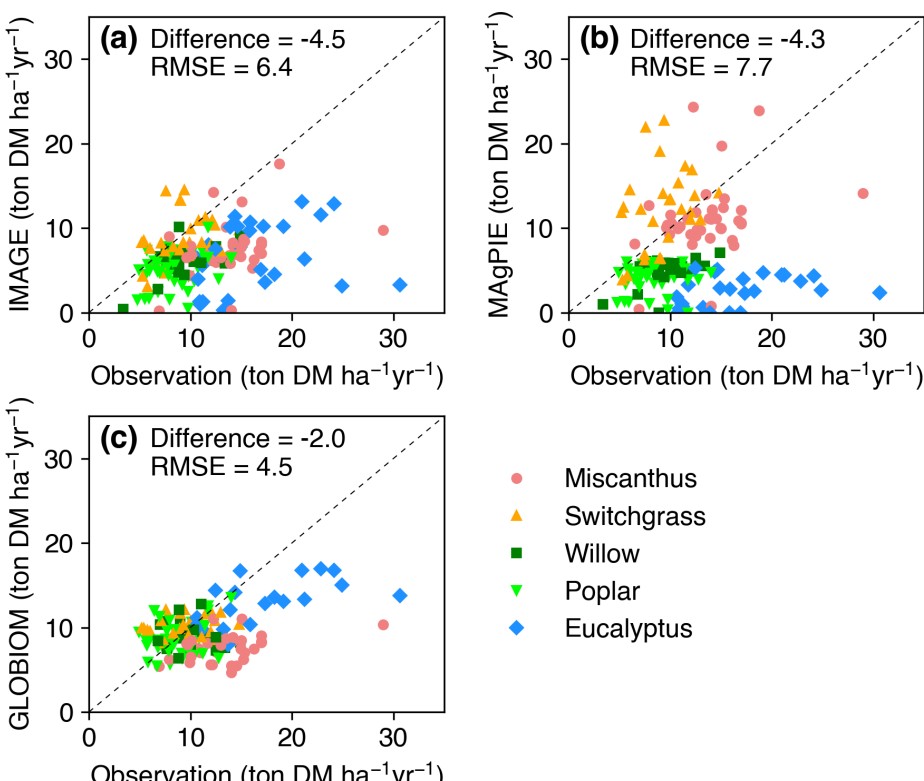

**Figure 6: Comparison of yields from random forest (RF) and IAM yield maps with site observations used to train the RF model (see the spatial distribution of sites in Fig. 1). Dash lines indicate the 1:1 lines. The median differences and root mean square errors (RMSE) between site observations and yields from RF and IAMs are also shown.**