# Peer review of "Mapping the yields of lignocellulosic bioenergy crops from observations at the global scale"

_Earth System Science Data, 2019_

## Referee Comment (RC1) · Anonymous Referee #1 · 27 Oct 2019

The authors use a machine learning technique (random forest-RF) to develop an up-scaled global (0.5 x 0.5 degrees) yield data set for five bioenergy crops. To justify how realistic this empirically-derived global bioenergy yield map, the authors further compare their product with the yield map used by the Integrated Assessment Models (IAM). In general, I agree with the authors that this dataset can become potentially a useful product for either benchmarking the global crop models (e.g. LPJ alike models) or being as input to IAMs. However, I think the method and results of this manuscript suffer from the following major weaknesses, which cannot make me convinced that this is a reliable product. 1. The authors disregard the details of temporal resolution and coverage of training data sets. 2. The authors haven't provided good reasoning for how they decided the training data sets. The temperature dataset in CRUNCEP is similar

to the CRU data set, which is based on observations, but precipitation has less good reliability. Also, why do authors choose satellite-based short-wave radiation? Does the median value in the high-resolution dataset have any advantages over the 0.5-degree data set (e.g. the CRU sunshine hours)? The water available index is a model-derived data set, but actually, there should be some satellite-based dataset to indicate soil moisture. In a word, I think the authors should give strong reasoning on why they have chosen their training data sets. 3. Given the big deviation shown between the yield map used by IAMs and the yield map derived by the authors, it is difficult to convince me of the reliability of the yield map generated by the random forest approach. I also wonder why the authors don't compare this product with their model estimates (Li et al., 2018b). Because the ORCHIDEE model has also been calibrated based on the same global bioenergy crop yield data set in Li et al. (2018a), it would be more logical to compare the derived product with the ORCHIDEE model estimate in the spatial scale.

---

## Referee Comment (RC2) · Anonymous Referee #2 · 28 Oct 2019

The authors reported 3,963 observations covering five bioenergy crops in the abstract, however, they only used 161 grid cells to train the RF model. The sample size is too limited to map the spatial distribution of global bioenergy crops (over 60,000 grid cells). The comparison of the derived maps with other modeled maps cannot convince me.

1. There were a bunch of variables included in the RF regressions. I suggested to add a diagram to show how random forest algorithm works in your study.

2. At the global scale, there are more than 60,000 grids in 0.5° × 0.5°. Here the authors used 161 grid cells for model training, among which you included five types of crop types. I think the training data are not substantial enough to build RF regression models.

[Figure]

3. Section 2.3. I appreciate that the authors compared their derived yield maps with the current three IAMs. However, it still cannot convince me since all these are modeled maps rather than the actual yield data. Is it possible to compare your derived yield maps with the existing inventory? Moreover, the authors assumed the derived maps are in 2010 without no temporal changes. To the best of my knowledge, the technology improvement has led to a significant increase of crop yield during the past several decades. Thus, I think it is not appropriate to compare your yield map with the present day's maps. The long-term average covering the time period of your collected observations is better for comparison. Line 198-199: What do you mean 'actual yield maps'? Is it your derived yield map from RF or other? If yes, I do not think you can consider it as an 'actual yield map'.

4. Figure 3. The spatial distribution of predicted yields seems to highly correlated with MAP. For example, the Amazon basin and Southeast Asia receive a substantial rainfall per year. The spatial distribution of Eucalypt and Miscanthus are so similar, the same as the remaining three crops. Thus, nothing new surprised me.

5. Figure 5. Did you compare your areas with any existing inventory data? It is better to compare yours with them since the total amount of production is also important.

6. Figure 2. You listed the variable importance in the trained RF model. It turns out that MAP is the dominant variable. You provide Figure S8 to show the relationship of bioenergy crop yield with temperature. However, MAT is not quite important compared with other variables. Why did not you show the relationship of each crop with dominant variables, such as MAP, GSL, WAI, etc.

---

## Author Comment (AC1) · 31 Dec 2019

**Response to comments**

**Paper #:** *essd-2019-118*
**Title:** *Mapping the yields of lignocellulosic bioenergy crops from observations at the global scale*
**Journal:** *Earth System Science Data*

**Reviewer #1:**

**Comment #1**

The authors use a machine learning technique (random forest-RF) to develop an upscaled global (0.5 x 0.5 degrees) yield data set for five bioenergy crops. To justify how realistic this empirically-derived global bioenergy yield map, the authors further compare their product with the yield map used by the Integrated Assessment Models (IAM). In general, I agree with the authors that this dataset can become potentially a useful product for either benchmarking the global crop models (e.g. LPJ alike models) or being as input to IAMs. However, I think the method and results of this manuscript suffer from the following major weaknesses, which cannot make me convinced that this is a reliable product.

**Response #1**

We thank the reviewer for the comments and suggestions. Please see the detailed point-by-point responses below.

**Comment #2**

1. The authors disregard the details of temporal resolution and coverage of training data sets.

**Response #2**

We agree that the temporal resolution and coverage of the training dataset are important for training the machine learning model given the temporal variations of climate conditions. Therefore, as suggested, we analyzed the sampling time in the training dataset. There are ~30% of the yield observations without reported sampling year in the original dataset and also ~30% in the aggregated 0.5-degree data used for random forest training. We thus arbitrarily set the 2 years before the publication year as the sampling year for the yield observations without reported sampling years (e.g. set 1997 as the sampling year if the reference paper was published in 1999). The frequency of the sampling years in the 0.5-degree data used for random forest training is shown in **Fig. R1**. The sampling years range from 1969 to 2016 with a median year of 1999.

We then derived temperature (T), precipitation (P) and short-wave radiation (SW, from CRUNCEP because BESS SW starts from 2001) and soil water availability index (WAI) at the sampling year for each grid cell and re-trained the random forest (RF). However, the OOB $R^2$ is **0.54**, **lower than** the original value of **0.63**. Possible reasons may include: 1) RF training may largely respond to the spatial gradients of climate and soil conditions, and thus the contribution of temporal variation may be low; 2) Climate conditions at the sampling year may be a good predictor of yields for annually harvested herbaceous crops, but yields of woody crops like eucalypt, poplar and willow may also be impacted by the previous years in the whole growing cycle. Unfortunately, there are only about 18% observations with both reported harvest year and age, impeding the derivation of the mean climate conditions during the whole growing cycle.

In addition, using the climate conditions at the sampling years also changed the variable importance (**Fig. R2**) compared to the original one (**Fig. 2a**). Precipitation is no longer an important contributing variable while contributions of the other variables are more or less similar to those in the original trained RF.

We will add this test as a sensitivity test and discuss accordingly in the revised manuscript.

**Figure R1 Histogram of sampling years in the yield observation grid data used for random forest training.**

[Figure]

**Figure R2 Variable importance in the trained RF model using climate conditions at the sampling years.**

[Figure]
* * *
**Comment #3**

2. The authors haven't provided good reasoning for how they decided the training data sets. The temperature dataset in CRUNCEP is similar to the CRU data set, which is based on observations, but precipitation has less good reliability. Also, why do authors choose satellite-based short-wave radiation? Does the median value in the high-resolution dataset have any advantages over the 0.5-degree data set (e.g. the CRU sunshine hours)? The water available index is a model-derived data set, but actually, there should be some satellite-based dataset to indicate soil moisture. In a word, I think the authors should give strong reasoning on why they have chosen their training data sets.

**Response #3**

We understand the reviewers' concerns, and we will add the reasons as well as more sensitivity tests to explain why we choose these climate forcing data in the revised manuscript (see below).

1) The CRUNCEP data is based on CRU climatology but only used NCEP to generate the diurnal and daily variability (*Viovy, 2017*; ftp://nacp.ornl.gov/synthesis/2009/frescati/model_driver/cru_ncep/analysis/readme.htm). We used **annual** precipitation in the random forest regression, and thus it should be the same as that from CRU.

2) In fact, the high-resolution datasets didn't help much in improving the Random Forest training. As discussed on **L313-325**, we tried higher-resolution (0.01 degree) MAP and MAT data from

WorldClim and trained the RF at higher resolution (0.01 degree) but the OOB $R^2$ didn't improve. We will further emphasize this point in the revised manuscript

As for the radiation data, shortwave radiation (SR) from CRUNCEP was simply converted from the cloudiness provided by CRU based on the calculation of clear sky incoming solar radiation as a function of date and latitude of each pixel (*Viovy, 2017*). By contrast, SR data from BESS was computed based on a series of forcing data from Terra & Aqua/MODIS Atmosphere and Land products, including "solar zenith angle from MODIS Atmospheric Profile product (MOD/MYD07_L2), dark target and deep blue combined aerosol optical depth at 500 nm from MODIS Aerosol product (MOD04_L2), cloud optical thickness, cloud top pressure, cloud top temperature, surface pressure and surface temperature from MODIS Cloud product (MOD06_L2), total column precipitable water vapor and total ozone burden from MODIS Atmospheric Profiles product (MOD/MYD07_L2), and land surface shortwave albedo from MODIS Albedo product (MCD43D61)" (*Ryu et al., 2018*). The SR data from BESS was also highly consistent with the observational field data ($R^2$=0.95, *Fig. 2* in *Ryu et al., 2018*). Therefore, we would expect SR from BESS is more reliable and accurate than SR from CRUNCEP. Still, we tested the RF performance using SR from CRUNCEP and the OOB $R^2$ remained unchanged (0.63), possibly due to the relatively low contribution of SR in the random forest training (7%, **Fig. 2a**) and the high spatial correlation between SR from BESS and from CRUNCEP. This will be added in the revised manuscript.

3) As suggested, we replaced the model-derived WAI with satellite-based surface soil moisture (SM) data, including the mean annual soil moisture data from Soil Moisture and Ocean Salinity (SMOS) during 2010-2018 (*Li et al., 2020*) and Soil Moisture Active Passive (SMAP) during 2015-2018, *O'Neill et al., 2019*). The OOB $R^2$ for SMOS and SMAP are 0.60 and 0.59 respectively, compared to the original value of 0.63. The lower performance may be caused by the fact that satellite-based soil moisture data only accounted for soil water status in the top centimeters whereas productivity is influenced by root-zone soil moisture. In addition, the importance ranking changed from #4 for WAI (**Fig. 2a**) to #8 for SM_SMOS and SM_SMAP (**Fig. R3**). The order of other variables remains unchanged. This will be added in the revised manuscript.

**Figure R3 Variable importance in the trained RF model using soil moisture (SM) data from SMOS (a) and SMAP (b).**

[Figure]

**Reference:**

*Li, X., Al-Yaari, A., Schwank, M., Fan, L., Frappart, F., Swenson, J., & Wigneron, J. P. Compared performances of SMOS-IC soil moisture and vegetation optical depth retrievals based on Tau-Omega and Two-Stream microwave emission models. Remote Sensing of Environment, 236, 111502. 2020.*

*O'Neill, P. E., S. Chan, E. G. Njoku, T. Jackson, and R. Bindlish. SMAP L3 Radiometer Global Daily 36 km EASE-Grid Soil Moisture, Version 6. [Indicate subset used]. Boulder, Colorado USA. NASA National Snow and Ice Data Center Distributed Active Archive Center. doi: https://doi.org/10.5067/EVYDQ32FNWTH. 2019.*

*Ryu, Y., Jiang, C., Kobayashi, H. and Detto, M.: MODIS-derived global land products of shortwave radiation and diffuse and total photosynthetically active radiation at 5 km resolution from 2000, Remote Sens. Environ., 204, 812–825, doi:10.1016/j.rse.2017.09.021, 2018.*

*Viovy, N. CRUNCEP dataset, description available at: ftp://nacp.ornl.gov/synthesis/2009/frescati/temp/land_use_change/original/readme.htm. 2017.*
* * *
**Comment #4**

3. Given the big deviation shown between the yield map used by IAMs and the yield map derived by the authors, it is difficult to convince me of the reliability of the yield map generated by the random forest approach. I also wonder why the authors don't compare this product with their model estimates (Li et al., 2018b). Because the ORCHIDEE model has also been calibrated based on the same global bioenergy crop yield data set in Li et al. (2018a), it would be more logical to compare the derived product with the ORCHIDEE model estimate in the spatial scale.

**Response #4**

As suggested, we compared the yield map derived from random forest with the yields simulated by the land surface model — ORCHIDEE (**Fig. R4**). Because poplar and willow were taken as one plant functional type (PFT) in ORCHIDEE, the average yields of poplar and willow from random forest were used for comparison (**Fig. R4b**). The yields simulated by ORCHIDEE are generally higher than those from random forest, especially for Miscanthus and Poplar&willow. This could be largely expected because in this version of ORCHIDEE, there are no nutrient limitations on plant growth, no effect of pests and disease on crops, and the management practices were implicitly included when adjusting the productivity parameters in the model to match the site observations with management like irrigation, fertilization or specific high-productive genotype. There could be a similar case in LPJml (*Heck et al., 2016*), and that is why the IAMs calibrated the LPJml yields based on currently observed yields to get the potential yield maps (see details **on L199-215**).

On the other hand, the predictions from random forest are largely constrained by the yield range of observations, representing the yields that can be achieved (or were achieved during the period when yield data were reported) under current (optimal) technology. This is exactly the purpose of producing this data product in our study, which is observation-based and can be used to benchmark the yields simulated by land surface models or IAMs.

**Figure R4 Comparison of bioenergy crop yields between the RF map and maps simulated by ORCHIDEE (ORCHIDEE yields minus RF yields where yields are available in both paired maps).**

[Figure]

---

## Author Comment (AC2) · 31 Dec 2019

**Response to comments**

**Paper #:** *essd-2019-118*
**Title:** *Mapping the yields of lignocellulosic bioenergy crops from observations at the global scale*
**Journal:** *Earth System Science Data*

**Reviewer #2:**

**Comment #1**

The authors reported 3,963 observations covering five bioenergy crops in the abstract, however, they only used 161 grid cells to train the RF model. The sample size is too limited to map the spatial distribution of global bioenergy crops (over 60,000 grid cells). The comparison of the derived maps with other modeled maps cannot convince me.

**Response #1**

We thank the reviewer for the comments and suggestions. Please see the detailed point-by-point responses below. For the sample size, please see **Response #3** for details.

**Comment #2**

1. There were a bunch of variables included in the RF regressions. I suggested to add a diagram to show how random forest algorithm works in your study.

**Response #2**

We will add it as suggested (**Fig. R5**).

**Fig R5 Workflow of random forest training and predicting in this study. The abbreviations of input variables can be found in Table 1.**

[Figure]

**Comment #3**

2. At the global scale, there are more than 60,000 grids in 0.5◦ × 0.5◦. Here the authors used 161 grid cells for model training, among which you included five types of crop types. I think the training data are not substantial enough to build RF regression models.

**Response #3**

We agree that if we only look at the grid cell number, the training dataset covers about ~0.3% (161 / 60,000) of the global total grid cells. However, the spatial representativeness of the sample is more important when being used to upscale the whole population pattern. As shown in **Fig. S7** (reproduced

below as **Fig. R6**), our training sample (gray) covers **most ranges of climate and soil variables** in the regions that we predicted (pink), implying that our training data are representative of the global adequate regions for bioenergy crop growth and thus appropriate for up-scaling (see **L363-373**). In addition to the range, the distributions also match well between the training sample and the prediction region (**Fig. R6**). Although the distributions of shortwave radiation are different, the importance of this variable in the random forest (RF) model is low (7%, **Fig. 2a**).

In addition, to avoid possible biases induced by out-of-range prediction, we only limited our predictions in regions with MAT and MAP above the minimums in the training data (**Section 2.2.3**). Thus, this gives us 33,216 grid cells in the prediction regions (instead of >60,000 globally) and avoids biased predictions in regions that are beyond the capacity of our trained random forest model. We can also add a short discussion on the comparison of the "out-of-range" predictions with IAM maps in the revised manuscript if needed.

At last, we would like to emphasize that we systematically collected all the published bioenergy crop yield observations that we searched in several literature databases (*Li et al. 2018*), so it is impossible to include more grid cells (currently 273 half-degree cells, 161 after selecting, **L157-171**) as there are no more observations available. Using these data, the OOB $R^2$ that serves as an evaluation of the trained random forest is 0.63, implying the trained RF algorithm is acceptable for prediction.

We will further summarize and discuss these points in the revised manuscript.

**Fig. R6 (S7) Distributions of explanatory variables in the training data and in the regions that are adequate for bioenergy crop growth. The ranges of variables for each bioenergy crop type in the training data are also shown as lines with different colors.**

[Figure]

**Reference:**

*Li, W., Ciais, P., Makowski, D. and Peng, S.: A global yield dataset for major lignocellulosic bioenergy crops based on field measurements, Sci. Data, 5(180169), 2018.*

**Comment #4**

3. Section 2.3. I appreciate that the authors compared their derived yield maps with the current three IAMs. However, it still cannot convince me since all these are modeled maps rather than the actual yield data. Is it possible to compare your derived yield maps with the existing inventory? Moreover, the authors assumed the derived maps are in 2010 without no temporal changes. To the best of my knowledge, the technology improvement has led to a significant increase of crop yield during the past several decades. Thus, I think it is not appropriate to compare your yield map with the present day's maps. The long-term average covering the time period of your collected observations is better for comparison. Line 198-199: What do you mean 'actual yield maps'? Is it your derived yield map from RF or other? If yes, I do not think you can consider it as an 'actual yield map'.

**Response #4**

In **Section 2.3**, we compared our random forest derived yield maps with those used in IAMs because our yield maps are observation based and can be used a benchmark for the present-day yield maps used in IAMs. Please see **Response #6** for the comparison with inventory data.

We agree with the reviewer that technology improvement has led to yield increase during the past decades, and thus "the long-term average covering the time period of collected observations" is better for comparison. However, the plantation of bioenergy crops applied in the IAMs is mainly for climate mitigation for removing $CO_2$ from the atmosphere e.g. through BECCS. This mitigation option has been proposed in most IAMs to keep the future temperature increase below 1.5 or 2 °C (*Rogelj et al., 2018*) but **not yet** implemented in large scales. Therefore, there are very limited (no) existing inventory data like e.g. those reported to the FAO by countries for other crops (see also **Response #6**), and the maps from IAMs start from present day. That is, unfortunately, **no** "long-term average covering the time period of collected observations" is available for comparison.

In addition, the comparison of our derived maps with maps from IAMs could be also justified: 1) the yield maps used in IMAGE and MAgPIE are from the simulated maps from LPJml model. In the model parameterization and calibration for bioenergy crops, LPJml also used available observation data (though a much smaller dataset compared to our dataset) covering the past period (e.g. at least since 1996 in *Beringer et al., 2011;* 1993-2008 in *Heck et al., 2016*). 2) The yield map from GLOBIOM is also based on historical observation data from FAO and other databases between 1984 and 2006 (see details on **L216-223**).

**L198-199**: Yes, "actual yield maps" is the derived yield map from RF. We call "actual yield maps" because our derived maps are based on observations and represent the yield that can be achieved under current (optimal) technology. We will revise this sentence as "For comparison, we used the present day (2010) actual yield maps (derived from RF).".

**Reference:**

*Beringer, T., Lucht, W. and Schaphoff, S.: Bioenergy production potential of global biomass plantations under environmental and agricultural constraints, GCB Bioenergy, 3(4), 299–312, doi:10.1111/j.1757-1707.2010.01088.x, 2011*

*Heck, V., Gerten, D., Lucht, W. and Boysen, L. R.: Is extensive terrestrial carbon dioxide removal a "green" form of geoengineering? A global modelling study, Glob. Planet. Change, 137, 123–130, doi:10.1016/j.gloplacha.2015.12.008, 2016*

*Rogelj, J., Popp, A., Calvin, K. V., Luderer, G., Emmerling, J., Gernaat, D., Fujimori, S., Strefler, J., Hasegawa, T., Marangoni, G., Krey, V., Kriegler, E., Riahi, K., Van Vuuren, D. P., Doelman, J., Drouet, L., Edmonds, J., Fricko, O., Harmsen, M., Havlík, P., Humpenöder, F., Stehfest, E. and Tavoni, M.: Scenarios towards limiting global mean temperature increase below 1.5 °C, Nat. Clim. Chang., doi:10.1038/s41558-018-0091-3, 2018.*

**Comment #5**

4. Figure 3. The spatial distribution of predicted yields seems to highly correlated with MAP. For example, the Amazon basin and Southeast Asia receive a substantial rainfall per year. The spatial

distribution of Eucalypt and Miscanthus are so similar, the same as the remaining three crops. Thus, nothing new surprised me.

**Response #5**

Yes, MAP as the most important variable in the RF regression is exactly what we obtained from the model training, and thus the predictions largely depend on the spatial patterns of annual rainfall. This is consistent with previous studies that MAP is the main predictor of NPP across spatial gradients (*Knapp et al., 2017*). Although the general spatial patterns look similar, there are still differences caused by other factors than MAP. This could be partly reflected by the different occupying regions from different bioenergy crops in **Fig. 3g**. To address the reviewer's concern on the similarity, we further plotted the map of yield differences between eucalypt and *Miscanthus* and among the other three crops. As shown in **Fig R7**, there are substantial differences between the yields of eucalypt and *Miscanthus*. The higher yields of eucalypt than *Miscanthus* in South America, East US, central Africa and southeast Asia and lower yields in other regions (**Fig. R7a**) can also be reflected by the best crop type in **Fig. 3g**. Because the contribution of crop types (poplar, switchgrass and willow) is low the trained random forest algorithm (CT_poplar, CT_switchgrass and CT_willow in **Fig. 2a**), the predicted yields in the regions where all three crops can grow are controlled by other mutual variables and thus similar. Therefore, the yield differences among these three crops are mainly caused by the different 'adequate' regions for growth (**Fig. S4**) defined by the minimum MAT and MAP in the observation dataset (**L181-190**). For example, willow can survive in regions with lower MAT and MAP, and thus have higher yield that poplar and switchgrass in these regions (**Fig. R7c,d**).

We will add the figure and corresponding discussion in the revised manuscript.

**Figure R7 Difference of predicted yields between various bioenergy crop types.**

[Figure]

**Reference:**

*Knapp, A. K., Ciais, P., & Smith, M. D. Reconciling inconsistencies in precipitation–productivity relationships: implications for climate change. New Phytologist, 214(1), 41-47, 2017.*

**Comment #6**

5. Figure 5. Did you compare your areas with any existing inventory data? It is better to compare yours with them since the total amount of production is also important.

**Response #6**

For Miscanthus and switchgrass, there are only small-scale experimental plots in different regions and no large-scale plantation, so, to the best of our knowledge, no region- or country-scale inventory data are available. Most yield data at farm levels were already included in our observation yield dataset (see "Field_type" and "Field_size" in *Table 2* in *Li et al. 2018*).

For poplar, willow and eucalypt, we searched on several literature databases and on Google but only found one *FAO* report by *Del Lungo et al.* (*FAO, 2006*). We collected the mean annual increment (MAI) data for species of *eucalyptus*, *populus* and *salix* for each country (**Table R1**, extracted from **Table 6a** in *FAO, 2006*). The volume unit of MAI was converted to mass unit of yield based on the wood density of different tree types (*Engineering ToolBox, 2004*).

The main difficulty is however lack of spatially explicit data about where are plantations located in national-scale inventory data, preventing an accurate comparison with the RF predicted yields. Still, we derived the yield range in the whole country from the RF predicted yield maps and compared with the yield range from the inventory data (*FAO, 2006*, **Fig. R8**). Most yield ranges from the inventory data overlapped with the ranges from RF maps (e.g. eucalypt and willow in Argentina) although the former is generally lower than the latter (**Fig. R8**). The higher minimum and maximum yields from RF could be caused partly by the exclusion of regions with MAP and MAT below the minimums from the observation dataset (to avoid out-of-range prediction, see details on **L181-190**). Especially, in some large countries, the inventory data may have plantations in some harsh climate and soils (e.g. most eucalypt plantations distribute in drier areas in the South Brazil). However, we must note that it is not a fair comparison without knowing the exact plantation locations in each country.

If the reviewer knows some other data sources, we will appreciate if you could let us know and we will add them for comparison.

**Table R1** Plantation area and maximum and minimum MAI (mean annual increment) of eucalypt, poplar and willow from inventory data compiled in *FAO 2006*.

| Species | Area (1000 ha) | MAI min ($m^3$/ha/y) | MAI max ($m^3$/ha/y) | Country |
|---|---|---|---|---|
| *Eucalyptus grandis* | 335 | 21 | 27 | South Africa |
| *Eucalyptus nitens* | 231 | 19 | 26 | South Africa |
| *Eucalyptus spp.* | 473 | 8 | 21 | Sudan |
| *Populus spp.* | 3220 | 9 | 18 | China |
| *Eucalyptus spp.* | 2397 | 8 | 21 | China |
| *Eucalyptus spp.* | 4047 | 8 | 21 | Indonesia, Viet Nam, India |
| *Populus spp.* | 171 | 9 | 18 | India |
| *Populus spp.* | 84 | 9 | 18 | Belgium, Netherlands, Ukraine, Latvia |
| *Populus hybrids* | 83 | 16 | 21 | Italy |
| *Eucalyptus globulus* | 442 | 16 | 25 | Australia |
| *Eucalyptus nitens* | 35 | 19 | 26 | Australia |
| *Eucalyptus dunnii* | 18 | 16 | 18 | Australia |
| *Eucalyptus grandis* | 18 | 21 | 27 | Australia |
| *Eucalyptus pilularis* | 18 | 18 | 18 | Australia |
| *Eucalyptus regnans* | 18 | 18 | 20 | Australia |
| *Eucalyptus spp.* | 3678 | 8 | 21 | Brazil, Chile |
| *Eucalyptus grandis* | 99 | 21 | 27 | Argentina |
| *Populus spp.* | 31 | 9 | 18 | Brazil, Chile |
| *Salix alba* | 23 | 13 | 20 | Argentina |
| *Salix babylonica* | 23 | 20 | 25 | Argentina |

| | | | | |
|---|---|---|---|---|
| *Salix babylonica var. sacramenta* | 23 | 20 | 25 | Argentina |
| *Salix hibrids* | 23 | 20 | 25 | Argentina |

**Figure R8 Yield ranges from (limited) inventory data and our random forest maps at country levels. IVI stands for Indonesia, Viet Nam and India; BNUL stands for Belgium, Netherlands, Ukraine and Latvia.**

[Figure]

**Reference:**

*Engineering ToolBox. Density of Various Wood Species. [online] Available at: https://www.engineeringtoolbox.com/wood-density-d_40.html [Accessed 15/11/2019]. 2004.*

*Brown, S. Estimating biomass and biomass change of tropical forests: a primer (Vol. 134). Food & Agriculture Org. 1997.*

*FAO. Global planted forests thematic study: results and analysis, by A. Del Lungo, J. Ball and J. Carle. Planted Forests and Trees Working Paper 38. Rome (also available at www.fao.org/forestry/site/10368/en). 2006.*

*Li, W., Ciais, P., Makowski, D. and Peng, S.: A global yield dataset for major lignocellulosic bioenergy crops based on field measurements, Sci. Data, 5(180169), 2018.*

**Comment #7**

6. Figure 2. You listed the variable importance in the trained RF model. It turns out that MAP is the dominant variable. You provide Figure S8 to show the relationship of bioenergy crop yield with temperature. However, MAT is not quite important compared with other variables. Why did not you show the relationship of each crop with dominant variables, such as MAP, GSL, WAI, etc.

**Response #7**

We only plotted the relationship with MAT because temperature is a target variable of future global warming and we would like to show how the yield will change with temperature increase in the future. We agree that MAP is the dominant variable in the RF, but temperature related variables (GSL and MAT) also contribute significantly. As suggested, we will further add the relationships with the dominant variables (reproduced below).

**Figure R9 Relationship of bioenergy crop yield with mean annual precipitation (MAP) across all grid cells that are adequate for bioenergy crop growth.**

[Figure]

**Figure R10 Relationship of bioenergy crop yield with growing season length (GSL) across all grid cells that are adequate for bioenergy crop growth.**

[Figure]

**Figure R11 Relationship of bioenergy crop yield with soil water availability index (WAI) across all grid cells that are adequate for bioenergy crop growth.**

[Figure]

**Figure R12 Relationship of bioenergy crop yield with growing season integrated normalized difference vegetation index (NDVI) across all grid cells that are adequate for bioenergy crop growth.**

[Figure]

**Figure R13 Relationship of bioenergy crop yield with shortwave radiation (SR) across all grid cells that are adequate for bioenergy crop growth.**

[Figure]

**Figure R14 Relationship of bioenergy crop yield with clay fraction (CF) across all grid cells that are adequate for bioenergy crop growth.**

[Figure]